# Exploring the Effect of Sampling Frequency on Real-World Mobility, Sedentary Behaviour, Physical Activity and Sleep Outcomes Measured with Wearable Devices in Rheumatoid Arthritis: Feasibility, Usability and Practical Considerations

**DOI:** 10.3390/bioengineering12010018

**Published:** 2024-12-28

**Authors:** Javad Sarvestan, Kenneth F. Baker, Silvia Del Din

**Affiliations:** 1Translational and Clinical Research Institute, Faculty of Medical Sciences, Newcastle University, Newcastle upon Tyne NE1 7RU, UK; javad.sarvestan@newcastle.ac.uk (J.S.); kenneth.baker@newcastle.ac.uk (K.F.B.); 2National Institute for Health and Care Research (NIHR), Newcastle Biomedical Research Centre (BRC), Newcastle University and The Newcastle upon Tyne Hospitals NHS Foundation Trust, Newcastle upon Tyne NE1 7RU, UK; 3Rheumatology Department, The Newcastle upon Tyne Hospitals NHS Foundation Trust, Newcastle upon Tyne NE1 7RU, UK

**Keywords:** gait, signal processing, continuous monitoring, downsampling

## Abstract

Modern treat-to-target management of rheumatoid arthritis (RA) involves titration of drug therapy to achieve remission, requiring close monitoring of disease activity through frequent clinical assessments. Accelerometry offers a novel method for continuous remote monitoring of RA activity by capturing fluctuations in mobility, sedentary behaviours, physical activity and sleep patterns over prolonged periods without the expense, inconvenience and environmental impact of extra hospital visits. We aimed to (a) assess the feasibility, usability and acceptability of wearable devices in patients with active RA; (b) investigate the multivariate relationships within the dataset; and (c) explore the robustness of accelerometry outcomes to downsampling to facilitate future prolonged monitoring. Eleven people with active RA newly starting an arthritis drug completed clinical assessments at 4-week intervals for 12 weeks. Participants wore an Axivity AX6 wrist device (sampling frequency 100 Hz) for 7 days after each clinical assessment. Measures of macro gait (volume, pattern and variability), micro gait (pace, rhythm, variability, asymmetry and postural control of walking), sedentary behaviour (standing, sitting and lying) and physical activity (moderate to vigorous physical activity [MVPA], sustained inactive bouts [SIBs]) and sleep outcomes (sleep duration, wake up after sleep onset, number of awakenings) were recorded. Feasibility, usability and acceptability of wearable devices were assessed using Rabinovich’s questionnaire, principal component (PC) analysis was used to investigate the multivariate relationships within the dataset, and Bland–Altman plots (bias and Limits of Agreement) and Intraclass Correlation Coefficient (ICC) were used to test the robustness of outcomes sampled at 100 Hz versus downsampled at 50 Hz and 25 Hz. Wearable devices obtained high feasibility, usability and acceptability scores among participants. Macro gait outcomes and MVPA (first PC) and micro gait outcomes and number of SIBs (second PC) exhibited the strongest loadings, with these first two PCs accounting for 40% of the variance of the dataset. Furthermore, these device metrics were robust to downsampling, showing good to excellent agreements (ICC ≥ 0.75). We identified two main domains of mobility, physical activity and sleep outcomes of people with RA: micro gait outcomes plus MVPA and micro gait outcomes plus number of SIBs. Combined with the high usability and acceptability of wearable devices and the robustness of outcomes to downsampling, our real-world data supports the feasibility of accelerometry for prolonged remote monitoring of RA disease activity.

## 1. Introduction

Rheumatoid arthritis (RA) is a chronic, progressive immune-mediated inflammatory disease affecting up to 1% of adults worldwide, characterised by joint and systemic inflammation [1]. Subsequent joint damage, disability, and decreased health status and quality of life (QoL) underscore the importance of close monitoring of RA disease activity and escalation of drug therapy to achieve remission—a cornerstone of modern RA management known as “treat-to-target” [2,3]. However, traditional methods of measuring disease activity in RA depend on regular physical assessments by rheumatology clinicians, requiring frequent hospital clinic visits. As such, there is a need to remotely monitor individuals with RA in real-world settings rather than confining assessments solely to intermittent physician-administered assessments [4].

Prolonged real-world remote monitoring of people with RA could reduce hospital/clinic visits, financial burden and resource utilisation by both patients and healthcare providers and minimise travel-related environmental impact. Furthermore, remote monitoring may also enhance timely intervention and equalise the distribution of medical services for individuals residing in rural areas [5]. Methods to assess RA symptomatology using questionnaire-based patient-reported outcome measures (PROMs) have been developed to facilitate remote monitoring in RA, and have been validated against clinician-applied measures of disease activity. However, these PROMs are often retrospective and, by definition, can only provide a subjective assessment. In contrast, accelerometry provides a novel approach to provide an objective, prolonged, continuous and remote monitoring of disease activity with the potential to augment treat-to-target management of patients with RA.

Nonetheless, there are two main challenges to the use of wearable accelerometry devices for prolonged remote monitoring of people with RA in the clinic, namely technical limitations and patient acceptability. Wearable devices, as recorded by accelerometers, pose several technical issues, including (a) large data footprint, (b) device storage, and (c) battery life [6], all of which become increasingly challenging with extended monitoring periods. Higher sampling frequency (e.g., 100 Hz) provides increased fidelity of raw data, but for a limited period of time, given the above constraints. Conversely, lower sampling frequency (e.g., 50 Hz or 25 Hz) allows for a longer recording time but may impact raw data quality.

Second, despite a high level of acceptability that has been previously reported regarding the long-term usage of body-worn wearable devices in other cohorts [7], this has not been tested in people with RA for prolonged remote monitoring [8]. It is important to understand what people with RA think about wearable devices, whether they find them acceptable, what problems they face in using them, and how they feel about them [9]. Moreover, the effectiveness, efficiency and satisfaction (indicating feasibility and usability of the system) with wearable devices can provide valuable insight into how people with RA think about their applications in real-world settings [10]. To this effect, this study aimed to (a) report the feasibility, usability and acceptability of wrist-worn devices for remote monitoring in people with RA; (b) explore the multivariate relationships within the mobility, sedentary behaviour, physical activity and sleep outcomes among people with RA; (c) evaluate the impact of downsampling (50 Hz and 25 Hz vs. 100 Hz) on mobility, sedentary behaviour, physical activity and sleep outcomes, by assessing agreement and robustness between outcomes evaluated at different sampling frequencies.

## 2. Methods

### 2.1. Participants

Adult patients >18 years old were recruited from routine outpatient rheumatology clinics at the Freeman Hospital (The Newcastle upon Tyne Hospitals NHS Foundation Trust, United Kingdom). To be eligible to participate, patients must have had a diagnosis of RA according to the 1987 American College of Rheumatology [11] or 2010 ACR/European Alliance of Associations for Rheumatology (EULAR) [12] classification criteria, be able to walk at least 4 metres independently without walking aids, and be within 4 weeks prior and 8 weeks after commencement of a new disease-modifying anti-rheumatic drug (DMARD: either conventional synthetic, targeted synthetic, or biologic agent). The latter criterion was designed to recruit patients whose disease activity was likely to improve during the study period and thus provide an opportunity to measure accelerometry outcomes over a wider dynamic range of arthritis activity. Ability to read and write in English was also a requirement to enable completion of patient questionnaires. Patients who had a diagnosis of a movement disorder and pregnant women were excluded due to potential confounding effects on accelerometry measurements. All patients provided informed written consent prior to study enrolment. The study was approved by the National Health Service Research Ethics Service (North of Scotland Research Ethics Committee, reference 22/NS/0072).

### 2.2. Procedure and Instruments

Participants were reviewed in a face-to-face assessment by a consultant rheumatologist (KB) at weeks 0, 4, 8 and 12. Disease activity was assessed at each visit using the disease activity score in 28 joints (DAS28)—a widely used composite disease activity score incorporating tender and swollen joint count, patient global score, and laboratory measures of inflammation (C-reactive protein [CRP] and erythrocyte sedimentation rate [ESR], giving the DAS28-CRP and DAS28-ESR scores, respectively [13,14,15]. Participants then wore two waterproof inertial measurement units (AX6, Axivity, York, UK, 23 × 32.5 × 8.9 mm, 11 g) on the lower back (L5 level) and non-dominant wrist using a lightweight, silicone wristband (Axivity, York, 16 g) continuously for 7 days following each clinical assessment (Figure 1) [16,17]. Previous studies showed comparable cut points for identification of physical activity levels between dominant and non-dominant wrists [18]. The AX6 device recorded three-dimensional acceleration (range = ±8 g) and gyroscope (±2000 degrees per second) raw data with a sampling frequency of 100 Hz [19]. Throughout the 7-day monitoring period, data were logged and stored on the device’s internal memory (1024 Mb flash non-volatile). After the monitoring period, participants returned the devices via prepaid envelopes, and data were downloaded using Open Movement software (OMGUI, v1.0.0.43). Following the final week 12 assessment, all participants completed a usability questionnaire to express their satisfaction levels and share their opinions regarding the acceptability of the wearable devices [20]. Rabinovich’s questionnaire has two sections: Section A comprises 12 questions using a 5-point ordinal scale. Responses per question were then transformed into a numerical scale from “1” to “5” whereby “1” signified low acceptance and “5” high acceptance, and answers with “no opinion” were scored as 3. In section B, participants were asked to give the device a single score between 0 and 100, with opportunity to provide free-text further comments [10,21].

### 2.3. Data Analysis

Downloaded triaxial acceleration signals were analysed using the original sampling frequency (100 Hz) and downsampled to 50 Hz and 25 Hz for further analysis. We downsampled the signals from 100 Hz to 50 Hz and 25 Hz to reduce computational complexity and ensure compatibility with previous studies in this domain. These rates were chosen as they exceeded the Nyquist frequency for the signals of interest, ensuring no loss of relevant information. Signal fidelity was confirmed by comparing key metrics before and after downsampling [22]. To remove low-frequency noise, we analysed the frequency spectrum of the raw signals and identified noise components below 1 Hz. A high-pass Butterworth filter with a cutoff frequency of 0.5 Hz was applied to eliminate low-frequency noise while preserving the signal’s critical components. This approach ensured that noise artefacts did not interfere with our analysis. The filtering parameters were selected based on prior studies of biomechanical signals, and the effectiveness was verified by inspecting the signal before and after filtering [23,24].

#### 2.3.1. Mobility Outcomes from Lower Back Device

For the evaluation of digital mobility outcomes, validated algorithms were used to first identify prolonged periods of walking (walking bouts) [25] and within each walking bout temporal events (foot initial and final contact events [26]) in order to quantify both “macro” gait outcomes (volume, pattern and variability of walking activity) and “micro” gait outcomes (discrete gait characteristics describing pace, rhythm, variability, asymmetry and postural control of walking). This algorithm, using only one IMU device placed on lower back, provides precise real-world gait macro and micro parameters while maintaining the minimum interference with either in- or outdoor daily activities for a continuous period of up to 10 days on the sampling frequency of 100 Hz [26,27].

Thus, step time was calculated as the time intervals between initial and final contact of one foot. Step length was calculated using inverted pendulum model applied to the centre of mass [27,28]. Thereafter, step velocity was calculated as the step length divided by step time. Variations in features, such as variability and asymmetry calculations, allow for a more precise analysis of step-by-step variations and inter-limb coordination, respectively [29]. The standard deviation between each step was used to measure variability, and asymmetry was calculated as the absolute difference between left and right steps [27].

For macro gait outcomes, volume was quantified using total steps, total bouts, and total walking time per day. Pattern was assessed using “alpha”, which was calculated as the shape and power law distribution of walking bouts based on a logarithmic scale from their density and length [30]. Alpha denotes the distribution of overall walking duration, elucidating the relationship between extended and short walking intervals. Greater alpha values signify that the walking duration primarily consists of shorter intervals, whereas decreased alpha values indicate a prevalence of longer intervals in the duration. Variability (S2) was computed by assessing the intra-subject variability of bout length. A greater S2 value indicates a diverse pattern of walking activities, whereas a lower S2 value suggests a limited variety of walking activities, indicating reduced engagement in different activities and a propensity to repeat the same activity pattern [31]. Total steps, total walk time and total bouts were calculated from the calculated steps.

#### 2.3.2. Sedentary Behaviour Outcomes from Lower Back Device

Sedentary behaviours, including standing, sitting and lying durations, were calculated according to the lower back device position and angles using previously validated algorithms by Abdul Jabbar, Sarvestan Abdul Jabbar, Sarvestan [32]. In this study, we considered 7 a.m. to 9 p.m. as daytime and 9 p.m. to 7 a.m. as night-time [33].

#### 2.3.3. Sleep and Activity Outcomes from Wristband Device

Using the open source and validated GGIR package (V2.8-0) [https://www.accelting.com/updates/ggir-release-2-8-0/, accessed on 11 October 2023], raw acceleration signals of all sampling frequencies were analysed [34]. In GGIR, signal processing involved self-calibration using local gravity and spotting prolonged high values [35]. It calculates the average dynamic acceleration magnitude, accounting for gravity, every 5 s and measures it in milligravitational units (mg). Non-wear time is identified if, for two axes out of three, the standard deviation within a 60 min period is under 13.0 mg, or the value range is less than 50 [36]. Valid wear time for one day was >16 h; data were excluded if participants wore the devices for less than 1 full day during the 7-day recording period [34]. For each sampling frequency, GGIR was used for quantifying measures of physical activity, including moderate-to-vigorous physical activity longer than 10 min (MVPA), as well as the duration and number of sustained inactivity bouts (SIB), along with sleep variables such as sleep duration and the number and duration of wake-ups after sleep onset (WASO) [37].

### 2.4. Statistical Analysis

#### 2.4.1. Data Exploration

Histograms and boxplots were visually inspected to assess the distribution of the data. Outliers (values that are 1.5 × interquartile range lower or greater than first or third quartiles, respectively) were identified but not excluded from analysis. Where appropriate, mean and standard deviation or median and range of the demographic and clinical characteristics were reported.

#### 2.4.2. Feasibility, Usability and Acceptability

The WHO report defines feasibility as “*whether the digital health system works as intended in a given context.*” [9]. To report feasibility, we assessed whether the intended data had been collected by each device (percentage of wristband and lower back device datasets collected over 7 days). Usability was also defined as “*whether the digital health system can be used as intended by users.*” [9]. To this effect, the usability of the devices was evaluated through analysis of the quantitative part of the usability questionnaire. Acceptability was evaluated through quantitative and qualitative (individual feedback) parts of the usability questionnaire [21].

#### 2.4.3. Outcome Selection

A principal component analysis (PCA) was conducted to identify the combination of features that most effectively represent digital mobility, sedentary behaviour, physical activity and sleep outcomes. Varimax rotation was utilised to calculate orthogonal factor scores, with a minimum eigenvalue for extraction set at 1. We analysed scree plots and factor loadings, which correspond to the quality of representation of variables in every dimension [38]. In selecting digital mobility, sedentary behaviour, and sleep outcomes for inclusion in the PCA, we aimed to incorporate a comprehensive set of variables that best describe the variance in the dataset. This approach aimed to accurately represent the underlying construct of each domain while preventing duplication and redundancy in the model. For instance, specific characteristics like step length were included, whereas similar variables within the same domain, such as stride length, were omitted to optimise model efficiency and avoid duplication and redundancy [39].

#### 2.4.4. Effect of Downsampling on Outcomes

To assess agreement between digital mobility, sedentary behaviours, physical activity and sleep outcomes evaluated using original (100 Hz) versus downsampled data (50 Hz and 25 Hz), Bland–Altman plots were used to quantify absolute agreement, including evaluation of bias and 95% Limits of Agreement (LoA). Next, relative errors were calculated between digital outcomes evaluated at different sampling frequencies. Finally, Intra-class Correlation Coefficients (ICCs (2,1)) were calculated to assess associations between digital outcomes [40]. Based on ICC estimates, values less than 0.5, between 0.5 and 0.75, between 0.75 and 0.9, and greater than 0.9 were deemed to be indicative of poor, moderate, good, and excellent agreement, respectively [41]. The impact of downsampling on digital outcomes was also assessed using parametric t-tests or Mann–Whitney U tests (if not normally distributed) test with a threshold of *p* < 0.05. All data and statistical analyses were conducted using MATLAB^®^ and Python.

## 3. Results

No statistically significant difference was found between any outcomes evaluated at the various sampling frequencies (minimum *p* = 0.39).

### 3.1. Patient Characteristics

Twelve participants were recruited for the study. One participant withdrew after the first visit following a lower limb fracture, and data from this participant were excluded from the analysis. The demographic and clinical data from the 11 participants included in the analysis are shown in Table 1.

### 3.2. Feasibility, Usability and Acceptability of Wearable Devices

The wristband device was fully worn throughout 7 days in all 44 recording periods (11 participants, 4 visits each) (Figure 2-a). The lower back device was consistently worn during all but one of the recording periods, when it was removed on the fourth day. With a total monitoring period of 100% for wristband devices and 99% for lower back devices, the wearable devices demonstrated robust feasibility among these participants with RA. Usability data were also good to excellent, with six participants rating the lower back and wristband devices with an overall usability score exceeding 95%, three participants rating above 90%, one participant rating above 85%, and another participant rating above 75% (Figure 2-b).

Participants had no trouble getting started with the devices (Q1), 73% found devices easy to put on/take off (Q2), while 55% experienced a little to moderate technical issues (Q3) (Figure 2-c). A total of 82% mentioned that devices did not interfere with their daily activities (Q4), and 73% reported that it was comfortable wearing them (Q5). A total of 91% did not feel embarrassed wearing the devices (Q6), and 82% found the instructions clear and working with the devices easy (Q7 and 8). A total of 91% of participants indicated that the devices were not bulky or heavy (Q9), did not bother them in bed (Q10), and did not feel their privacy was affected by the devices (Q11). A total of 83% expressed willingness to wear devices for more than one week if their doctor asked them to (Q12). Regarding the acceptability of the wearable devices, two participants noted that the lower back device interfered with their bath and swimming, and two others felt uncomfortable as it was in the waistband area. Regarding the wrist device, two participants mentioned that there could be an improvement in the quality of the rubber band material used. Nevertheless, all participants unanimously mentioned that both devices were easy and discrete to wear.

### 3.3. Quality of Representation of Digital Mobility, Sedentary Behaviours, Physical Activity and Sleep Outcomes

The percentage of explained variance for all PCs is presented in the scree plot (Figure 3-a). The first two PCs contribute to nearly 40% of the total variance of the dataset. From the lower back device variables in the first PC, walk time per day (0.38), steps per day (0.37) and alpha (0.32) demonstrated the highest cos2 values, while the MVPA (0.36) was the only variable from wristband device with comparable cos2 in the first PC (Figure 3-b). In the second dimension (PC2), micro gait outcomes portrayed the highest quality of representation, including time variability (0.41) and step velocity (0.37), as well as the number of SIBs from the wristband variables.

A two-dimensional (PC1 and PC2) loading of the entire dataset is presented in Figure 3-c. In the first dimension (PC1), the volume of macro gait outcomes (walk time per day and steps per day) and MVPA were aligned together and opposite to step length asymmetry (postural control). The second dimension portrays the alignment of step time and step time variability, which were opposite in direction to step length and step velocity. The number of SIBs and SIB duration, on the other hand, were in the same direction and opposite to bouts per day. Alpha (pattern) was also opposite to variability, sleep duration and lying volumes during day and night.

### 3.4. Agreement Between Mobility, Sedentary Behaviour, Physical Activity and Sleep Outcomes Quantified at Different Sampling Frequencies

Descriptive measures, bias, Limits of Agreement (LoA), relative error and Intraclass Correlation Coefficient (ICC) values for mobility, sedentary behaviour, physical activity and sleep outcomes between original (100 Hz) versus downsampled data (50 Hz and 25 Hz) are presented in Table 2 and Figure 4. The entire series of Bland–Altman graphs are provided in Appendix A. In terms of macro gait outcomes, good to excellent intra-class correlation coefficient (ICC) values (ranging from 0.75 to 0.99) were observed for either 50 Hz vs. 100 Hz or 25 Hz vs. 100 Hz. This was linked with low relative errors (maximum relative error = 3.94%) and relatively low Limits of Agreement (LoAs) among all macro gait outcomes. Similarly, for micro gait outcomes, excellent ICC values were observed for the entire variables (minimum ICC = 0.90), with low relative errors (maximum relative error = 3.55%) and LoAs. An excellent ICC value of 0.99 was observed for the entire sedentary behaviour outcomes, with a maximum relative error of 5.34% and a maximum bias of 0.06 h for sitting volume during the day. Among wristband device outcomes, sleep duration showed an excellent ICC of 1.00 for both 50 Hz vs. 100 Hz and 25 Hz vs. 100 Hz (maximum relative error = 0.10). All other outcomes exhibited excellent ICC of >0.92 for both sampling frequencies (maximum relative error = 20.82%).

## 4. Discussion

This study aimed to (a) assess the feasibility, usability and acceptability of wearable devices among patients with RA, (b) investigate the multivariate relationships within the dataset, and (c) explore the robustness of outcomes to downsampling. Accelerometry obtained high feasibility, usability and acceptability scores among participants. Macro gait outcomes and MVPA portrayed the highest loadings in the first PC, while micro gait outcomes and the number of SIBs exhibited the highest loadings in the second PC. Furthermore, accelerometer metrics were robust to downsampling, showing good to excellent agreements.

### 4.1. Feasibility, Usability and Acceptability of Wearable Devices

Rabinovich’s usability questionnaire is widely used for the evaluation of the feasibility, usability and acceptability of wearable devices among different cohorts [7]. According to part A of this questionnaire (3rd question), 45% of participants found wearable devices “*most favourable*”, while 36% found it “*favourable*” and 18% had a neutral opinion about its feasibility. This was associated with 100% and 99% worn time throughout 7 days for wristband and lower back devices, respectively. Thus, the outcomes of our study highlight that the devices were both acceptable and feasible among people with RA. The entire participants rated the usability questions (1st, 2nd, 7th and 8th) as “favourable” (27%) or “most favourable” (73%). Furthermore, according to section B of the questionnaire, excepting one participant who found the wristband device too bulky and the lower back device interfering with taking a bath, the rest of the participants provided a usability score of above 85%, indicating that the wearable devices can be used as intended by the vast majority of users. These findings align with previous research advocating for the customisation of wearable devices according to individuals’ needs and motor symptoms [42]. Indeed, superior feasibility and usability scores have been observed for patients with RA compared to those with Parkinson’s disease, multiple sclerosis, congestive heart failure and chronic obstructive pulmonary disease [10,21], suggesting greater potential for the application of wearable devices among patients with RA. Nevertheless, due to the small sample size in this study, the wearability of IMU devices should be further explored in a broader population to gain a deeper understanding of user needs and preferences for these devices.

### 4.2. Quality of Representation of Digital Mobility, Sedentary Behaviours and Sleep Outcomes

The PCA revealed that the mobility, sedentary behaviour, physical activity and sleep of people with RA could be classified into two main domains. The first domain consists mainly of macro gait outcomes and MVPA, indicating that this dimension places its emphasis on the larger aspect of physical activity, including the distribution of patterns of walking bouts (alpha), walk time per day and steps per day. The second domain encompasses step time and its variability, step velocity and step length from micro gait outcomes, and the number of SIBs from physical activity, indicating that this dimension mainly emphasises the rhythm and pace from gait and inactive bouts during the day.

The two-dimensional QoR of variables gives insight into the multidimensional nature of mobility, sedentary behaviour, physical activity and sleep among people with RA, with the opposing directionality of opposing outcome measures lending some face validity to the dataset. For example, in the first dimension (PC1), steps per day and walk time per day were in the opposite direction to sitting volume during the day, and activity bouts per day were in the opposite direction to the number and duration of SIBs. Among micro gait outcomes (PC2), step velocity and length were in the opposite direction of step time. Furthermore, we observed a good alignment of outcomes derived from wristbands versus lower back devices. For instance, MVPA (wristband) was aligned with walk time per day and total steps (lower back), and sleep duration (wristband) was aligned with lying volume during the night (lower back). This suggests that different monitoring sites provide similar insights into individuals’ mobility, sedentary behaviour, physical activity and sleep.

### 4.3. Robustness and Agreement Between Digital Mobility, Sedentary Behaviours and Sleep Outcomes Quantified at Different Sampling Frequencies

There are few studies focused on the impacts of downsampling on GGIR outcomes, but they mostly explored the technical aspects (i.e., Euclidian norm minus one) [43], or general physical activity or sleep duration [44]. Furthermore, this study is the first study that explored the effect of downsampling on the outcomes driven by lower back devices (mobility and sedentary behaviour). Among the lower back-driven outcomes, the alpha values exhibited the lowest ICC (0.75) when comparing 25 Hz to 100 Hz. Despite this, the ICC still falls within the “good” range (relative error of 3.58%), indicating a reasonably strong correlation. The entire macro and micro gait and sedentary behaviour outcomes revealed “good” to “excellent” ICC values, indicating their robustness to the downsampling. Moreover, the GGIR package portrayed excellent reliability (minimum ICC = 0.92) when comparing wristband-driven outcomes at 50 Hz and 25 Hz vs. 100 Hz. Nevertheless, a relative error of 20.82% for the number of SIBs between 25 Hz vs. 100 Hz might suggest that the ICC value of 0.95 might overestimate its performance in the identification of SIBs at lower sampling frequencies. Nevertheless, downsampling to 25 Hz would reduce data file size 4-fold, thus prolonging recording capacity to 4 weeks rather than 1 week, allowing extended monitoring. This underscores the potential benefits of adjusting sampling frequencies to optimise data collection and storage in wearable device applications for people with RA. In addition to reducing the data footprint, it streamlines data postprocessing and enables faster delivery [45].

### 4.4. Limitations

The study results must be interpreted alongside its limitations. First, the small number of participants limits generalisability to the broader population and prevents further subgroup analyses. Second, considering that the majority of participants were in the early stages of the disease, further studies are required to observe individuals with more established RA. Third, the wearability of IMU devices should be further explored in a broader population to gain a deeper understanding of user needs and preferences for these devices. Finally, the observed effects of data downsampling may not apply to different wearable device manufacturers (i.e., APDM, ActiGraph).

## 5. Conclusions

Our study supports three key findings: first, it is feasible to use wearable devices to track the mobility, sedentary behaviour, physical activity and sleep of people with RA in real-world settings. Second, PCA uncovers two primary domains for the characterisation of mobility and physical activity: the first domain encompasses macro gait outcomes and MVPA categorisation, while the second domain comprises micro gait outcomes along with the number and duration of SIBs. Third, the robustness of mobility, sedentary behaviour, physical activity and sleep outcomes to downsampling supports the feasibility of extended monitoring of individuals with RA for application in future clinical trials.

## Figures and Tables

**Figure 1 bioengineering-12-00018-f001:**
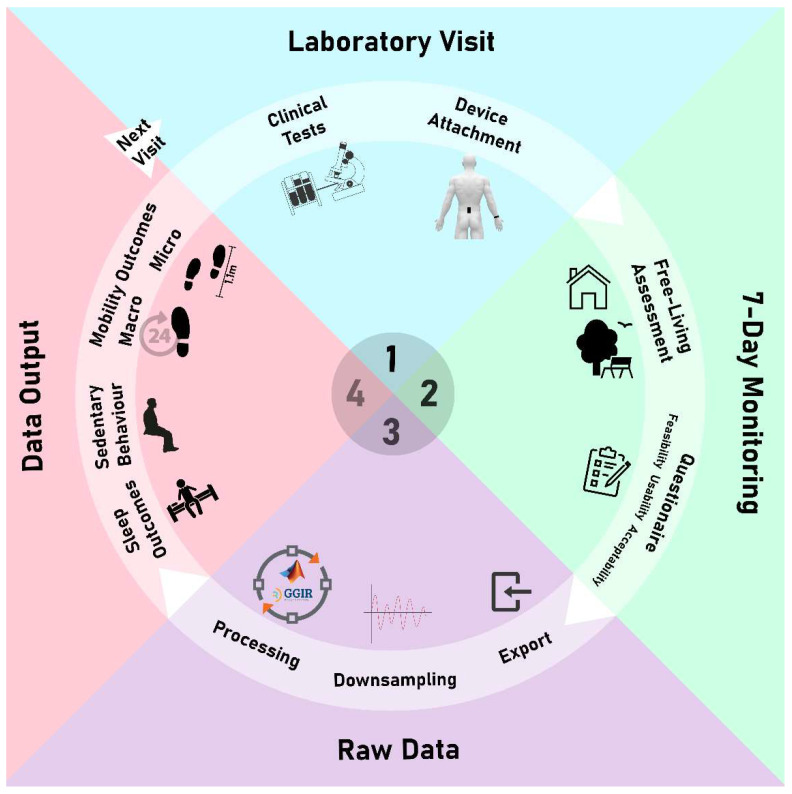
Study protocol: phase 1—laboratory visit including clinical tests and device attachment; phase 2—7-day monitoring including real-world assessment using lower back and wrist devices and feasibility, usability and acceptability questionnaires; phase 3—raw data management including data export, downsampling and processing; and phase 4—data analysis including sleep, sedentary behaviour, and digital mobility outcomes.

**Figure 2 bioengineering-12-00018-f002:**
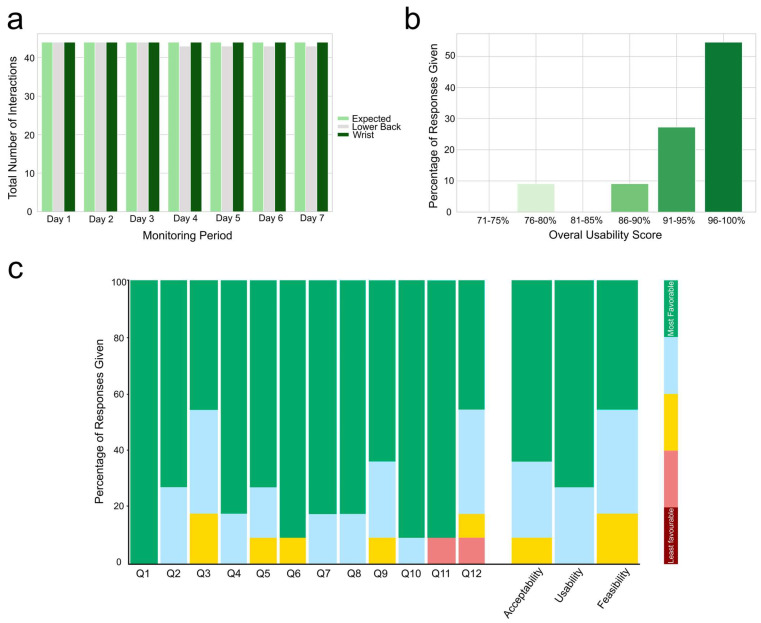
(**a**) Interactions with lower back and wristband devices throughout 44 assessments (11 participants × 4 time points). (**b**) Overall usability score to lower back and wristband devices. (**c**) Responses given to the usability questionnaire (%). Q1: How much trouble did you have getting started with the wearable devices? Q2: The wearable devices were easy to put on/take off. Q3: I experienced technical problems with the wearable devices. Q4: The wearable devices interfered with my normal activities. Q5: I felt comfortable wearing the wearable devices. Q6: I felt embarrassed wearing the wearable devices. Q7: The instructions on how to use the wearable devices were clear. Q8: Using the wearable devices on a daily basis was easy. Q9: The wearable devices were bulky/heavy. Q10: The wearable devices bothered me in bed. Q11: I felt my privacy was invaded by the wearable devices. Q12: If my doctor would like to use the wearable devices to assess my arthritis, I would be willing to wear them and use them for [duration]. The usability questionnaire responses are categorised based on acceptability (Q4, Q5, Q6, Q9, Q10, Q11, and Q12), usability (Q1, Q2, Q7, and Q8), and feasibility (Q3) themes. Responses are colour-coded on a scale from red (score = 1 indicating the least favourable response) to green (score = 5 indicating the most favourable response). In this questionnaire, “wearable devices” pertains to the lower back and wrist devices [21].

**Figure 3 bioengineering-12-00018-f003:**
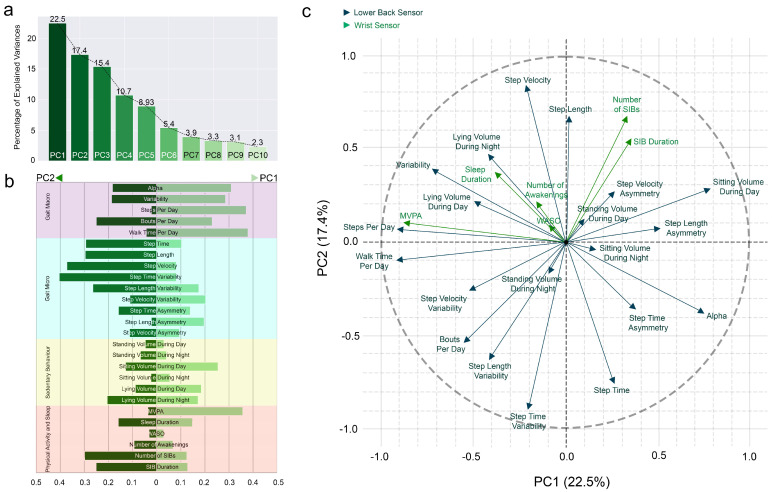
(**a**) Scree plot of percentage of explained variance by the first 10 principal components. (**b**,**c**) Quality of representation (**b**) and directionality (**c**) of all variables in the first two principal components.

**Figure 4 bioengineering-12-00018-f004:**
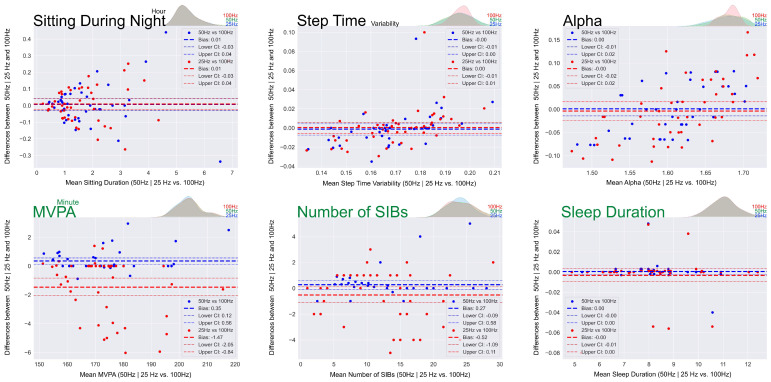
Bland–Altman and density plots of selected mobility, sedentary behaviour, physical activity and sleep outcomes from the lower back (black) and wrist (green) devices at 50 Hz and 25 Hz versus 100 Hz. Full sets of graphs are provided in the Appendix A.

**Table 1 bioengineering-12-00018-t001:** Demographic and clinical characteristics of the cohort.

Characteristic	Value
Female sex: n/N (%)	6/11 (55)
Age (years): median (IQR) [range]	75 (70–76.5) [47–90]
Height (m): median (IQR) [range]	1.70 (1.60–1.76) [1.54–1.82]
Weight (kg): median (IQR) [range]	72.8 (62.4–79.4) [57.5–97.0]
BMI: median (IQR) [range]	25.4 (24.3–26.7) [21.2–32.0]
Months since diagnosis: median (IQR) [range]	5 (2.5–15.5) [1–336]
Rheumatoid factor positive: n (%)	8 (73)
ACPA positive: n (%)	7 (64)
Rheumatoid factor and ACPA double positive: n (%)	7 (64)
Current DMARD therapy: n (%)	
Methotrexate	8 (64)
Sulfasalazine	4 (36)
Hydroxychloroquine	3 (27)
Leflunomide	2 (18)
Adalimumab	1 (9)

ACPA: anti-citrullinated peptide antibody; BMI: body mass index; DMARD: disease-modifying anti-rheumatic drug.

**Table 2 bioengineering-12-00018-t002:** Descriptive measures, bias, Limits of Agreement (LoA), relative error and Intraclass Correlation Coefficient (ICC) values for mobility, sedentary behaviour, physical activity and sleep outcomes between original (100 Hz) versus downsampled data (50 Hz and 25 Hz).

Domain	Variable	50 Hz vs. 100 Hz					25 Hz vs. 100 Hz			
		100 Hz Mean ± SD	50 Hz Mean ± SD	Bias [LoA]	Relative Error (%)	ICC [L95%–U95%]	25 Hz Mean ± SD	Bias [LoA]	Relative Error (%)	ICC [L95%–U95%]
**Mobility (Macro)**	Alpha	1.61 ± 0.05	1.62 ± 0.08	0.00 [−0.01, 0.02]	2.86	0.77 [0.57–0.90]	1.62 ± 0.09	0.00 [−0.02, 0.02]	3.58	0.75 [0.51–0.93]
	Variability (S2)	0.82 ± 0.08	0.83 ± 0.08	0.00 [−0.01, 0.01]	2.86	0.94 [0.90–0.97]	0.83 ± 0.08	0.00 [−0.01, 0.01]	3.94	0.90 [0.83–0.94]
	Steps per Day	12,559 ± 4886	12,525 ± 4857	34.11 [−74, 136]	2.28	0.99 [0.99–1.00]	12,485 ± 4836	−39.73 [−170, 94]	2.92	0.99 [0.99–1.00]
	Walk Time per Day (m)	181.9 ± 65	180.9 ± 63	−0.95 [−2.55, 0.65]	2.39	0.99 [0.99–1.00]	175.7 ± 63	−6.25 [−7.19, −5.17]	3.65	0.99 [0.99–1.00]
	Bouts per Day	662 ± 172	663 ± 174	1.48 [−4.11, 7.05]	2.42	0.99 [0.99–1.00]	665 ± 181	3.05 [−2.89, 8.96]	2.50	0.99 [0.99–1.00]
**Mobility (Micro)**	Step Time (s)	0.61 ± 0.03	0.60 ± 0.03	0.00 [−0.01, 0.00]	2.90	0.99 [0.99–1.00]	0.59 ± 0.03	−0.01 [−0.01, 0.00]	3.55	0.99 [0.99–1.00]
	Step Length (m)	0.60 ± 0.03	0.59 ± 0.03	0.00 [−0.01, 0.00]	2.68	0.91 [0.82–0.95]	0.59 ± 0.03	0.00 [−0.01, 0.00]	3.19	0.87 [0.77–0.93]
	Step Velocity (m·s^−1^)	1.06 ± 0.09	1.03 ± 0.09	−0.06 [−0.07, −0.05]	3.23	0.95 [0.91–0.97]	1.04 ± 0.09	−0.05 [−0.06, 0.04]	2.82	0.97 [0.94–0.98]
	Step Time Variability (s)	0.17 ± 0.01	0.17 ± 0.01	0.00 [−0.01, 0.00]	2.95	0.96 [0.93–0.98]	0.17 ± 0.01	0.00 [−0.01, 0.01]	2.90	0.97 [0.94–0.98]
	Step Length Variability (m)	0.15 ± 0.01	0.14 ± 0.01	0.00 [−0.00, 0.00]	2.75	0.93 [0.88–0.96]	0.14 ± 0.01	0.00 [0.00, 0.00]	3.13	0.92 [0.85–0.95]
	Step Velocity Variability (m·s^−1^)	0.36 ± 0.03	0.35 ± 0.04	0.00 [−0.00, 0.00]	3.19	0.96 [0.92–0.98]	0.35 ± 0.03	0.00 [0.00, 0.01]	3.02	0.96 [0.93–0.98]
	Step Time Asymmetry (s)	0.09 ± 0.01	0.09 ± 0.02	0.00 [−0.00, 0.00]	3.07	0.97 [0.95–0.99]	0.09 ± 0.02	0.00 [0.00, 0.00]	3.11	0.97 [0.95–0.98]
	Step Length Asymmetry (m)	0.09 ± 0.01	0.09 ± 0.01	0.00 [−0.00, 0.00]	3.17	0.97 [0.95–0.99]	0.09 ± 0.01	0.00 [0.00, 0.00]	3.21	0.97 [0.96–0.99]
	Step Velocity Asymmetry (m·s^−1^)	0.21 ± 0.03	0.20 ± 0.03	0.00 [−0.00, 0.00]	3.14	0.98 [0.97–0.99]	0.20 ± 0.03	0.00 [0.00, 0.00]	3.35	0.98 [0.96–0.99]
**Sedentary Behaviour**	Standing During Day (h)	0.42 ± 0.39	0.42 ± 0.39	0.00 [−0.01, 0.01]	1.90	0.99 [0.99–1.00]	0.44 ± 0.41	0.01 [0.00, 0.02]	2.02	0.99 [0.99–1.00]
	Standing During Night (h)	0.11 ± 0.10	0.11 ± 0.10	0.00 [−0.00, 0.00]	2.12	0.99 [0.99–1.00]	0.11 ± 0.11	0.00 [0.00, 0.00]	2.09	0.99 [0.99–1.00]
	Sitting During Day (h)	4.84 ± 1.91	4.84 ± 1.93	−0.01 [−0.11, 0.09]	5.34	0.99 [0.99–1.00]	4.90 ± 1.95	0.06 [−0.02, 0.14]	4.23	0.99 [0.99–1.00]
	Sitting During Night (h)	1.74 ± 1.22	1.74 ± 1.22	0.01 [−0.03, 0.04]	4.43	0.99 [0.99–1.00]	1.75 ± 1.26	0.01 [−0.03, 0.04]	5.50	0.99 [0.99–1.00]
	Lying During Day (h)	0.89 ± 0.69	0.90 ± 0.70	0.00 [−0.02, 0.02]	4.90	0.99 [0.99–1.00]	0.89 ± 0.67	−0.01 [−0.03, 0.01]	5.01	0.99 [0.99–1.00]
	Lying During Night (h)	7.05 ± 1.21	7.02 ± 1.25	−0.03 [−0.14, 0.07]	4.21	0.99 [0.99–1.00]	7.05 ± 1.23	0.00 [−0.11, 0.11]	4.43	0.99 [0.99–1.00]
**Physical activity and sleep**	MVPA (m)	173.7 ± 13.5	174.1 ± 13.6	0.35 [0.12, 0.56]	0.24	0.99 [0.99–1.00]	172.2 ± 12.7	−1.47 [−2.05, −0.84]	0.92	0.98 [0.97–0.99]
	Sleep Duration (h)	8.1 ± 1.7	8.1 ± 1.7	0.00 [0.00, 0.00]	0.03	1.00 [0.99–1.00]	8.1 ± 1.7	0.00 [−0.01, 0.00]	0.10	1.00 [0.99–1.00]
	WASO (m)	1.9 ± 1.9	1.8 ± 2.0	−0.01 [−0.06, 0.05]	2.76	0.99 [0.99–1.00]	1.9 ± 2.0	0.02 [−0.06, 0.09]	4.43	0.99 [0.99–1.00]
	Number of Awakenings	20.9 ± 7.7	20.7 ± 7.9	−0.21 [−1.01, 0.63]	11.12	0.94 [0.89–0.96]	21.1 ± 7.9	0.18 [−0.75, 1.11]	12.46	0.92 [0.86–0.96]
	SIB Duration (h)	1.1 ± 1.0	1.1 ± 1.1	0.01 [−0.06, 0.07]	2.15	0.98 [0.96–0.99]	1.1 ± 1.0	−0.12 [−0.23, 0.00]	3.32	0.94 [0.90–0.97]
	Number of SIBs	12.3 ± 6.7	12.5 ± 7.0	0.27 [−0.09, 0.58]	8.18	0.98 [0.97–0.99]	11.7 ± 6.2	−0.52 [−1.09, 0.11]	20.82	0.95 [0.90–0.97]

Alpha = distributions of walking bouts, Variability (S2) = variability of bout length, MVPA = moderate to vigorous physical activity, WASO = wake up after sleep onset duration, SIB = sustained inactivity bout, LoA = Limits of Agreement, ICC = Intraclass Correlation Coefficient. U95% = upper bounds of the 95% confidence interval, L95% = lower bounds of the 95% confidence interval.

## Data Availability

Data will be available on a formal request from corresponding author.

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
