# Peer review of "Exploring the Effect of Sampling Frequency on Real-World Mobility, Sedentary Behaviour, Physical Activity and Sleep Outcomes Measured with Wearable Devices in Rheumatoid Arthritis: Feasibility, Usability and Practical Considerations"

_bioengineering, 2024, doi:10.3390/bioengineering12010018_

Round 1
Reviewer 1 Report
Comments and Suggestions for Authors
Q1. What the main idea behind patient acceptability?
Q2. Disease activity was assessed- Clinical or mental.
Q3. Question types are not specified and how these are selected.
Q4. The requirement of down sampled is not specified.
Q5. For the evaluation of digital mobility outcomes existing algorithm is used without specifying its advantages.
Q6. How low frequency noise of data is eliminated?
Q7. Authors studied the disease of the patient age more than 45 years. The actual detection of disease is based on age i.e. 50 years and above. So it cannot be the limitation.
Comments on the Quality of English LanguageThere are many places where it can be simplified
Author Response
Dear respected reviewer,
Hello and thank you for your constructive comments. We tried to amend the manuscript in line with your comments and suggestions. Please find our responses below:
- What the main idea behind patient acceptability?
Assessing the acceptability of wearables among different cohorts, particularly people with RA, is crucial for the patient-specific design and application of these technologies. A usability, acceptability, and feasibility questionnaire would provide valuable insights for future studies, enabling the development of wearables that are better suited to the needs of people with RA while minimizing interference with their daily activities.
- Disease activity was assessed- Clinical or mental.
Clinical disease activity was assessed by a medically-trained consultant rheumatologist (KB) according to the disease activity score in 28 joints (DAS28). This is a widely used measure of rheumatoid arthritis disease activity both in clinical practice and research studies for the past 3 decades. The DAS28 is a composite score based on clinical examination findings (tender and swollen joint counts), a patient global assessment of their arthritis activity (on a scale of 0-100), and a laboratory measure of inflammation (either C-reactive protein or erythrocyte sedimentation rate) Please find the explanation in lines 112-117.
- Question types are not specified and how these are selected.
Thank you for your comment. We utilized Rabinovich’s questionnaire ("Validity of Physical Activity Monitors During Daily Life in Patients with COPD," 2013) to assess usability, feasibility, and acceptability (Lines 130-135). The specific questions, along with the criteria for classifying usability, feasibility, and acceptability, are detailed in the caption of Figure 2.
- The requirement of down sampled is not specified.
Thank you for noticing this. We provided the requirements for downsampling to the lines 144-149.
- For the evaluation of digital mobility outcomes existing algorithm is used without specifying its advantages.
Thanks for your suggestion, we added the advantages to the lines 162-165.
- How low frequency noise of data is eliminated?
Thanks for noticing the missing section. We added this information to the lines 149-155.
- Authors studied the disease of the patient age more than 45 years. The actual detection of disease is based on age i.e. 50 years and above. So, it cannot be the limitation.
Thank you for your suggestion. Rheumatoid arthritis can be diagnosed in any individual aged 16 years and above – however, in this study the age range of our participants was from 47-90 years, and thus lack of recruitment of younger patients is a limitation of our results. We have amended the wording of the limitations according to your comment.
Reviewer 2 Report
Comments and Suggestions for Authors
See attached file

Author Response
Dear respected reviewer,
Hello and thank you for your constructive comments. We tried to amend the manuscript in line with your comments and suggestions. Please find our responses below:
The authors of the work “Exploring the Effect of Sampling Frequency on Real-World Mobility, Sedentary Behaviour, Physical Activity and Sleep … ” studied the possibility of using wearable devices, containing accelerometers and gyroscopes, in patients with active rheumatoid arthritis (RA) to assess the feasibility of measured data, usability and acceptance by patients, as well as the possibility of recording data with lower sampling frequencies. One of the aims behind this study was to establish if measuring disease activity in RA patients could be done in real-world settings instead of through regular assessments by rheumatology clinicians, what would reduce clinic visits and costs. Only eleven adult patients with a diagnosis of RA that could walk at least four meters without walking aids participated in the study. They wore two recording devices, one in the wrist and the other on the lower back (L5 level). Measurements were performed in periods of seven days after each clinical assessment. Measurements of different parameters were recorded and analyzed using principal component techniques. Usability and acceptability were assessed by using Rabinovich’s questionnaires. As the authors state, the small number of participants, and the limitations of their ages, as well as their early stages of the disease, are two limitations of the study that would require further studies in order to obtain broader conclusions. The authors conclude that, according to their results, it is feasible to use wearable devices to track the activity of RA patients in real-world settings, and that PC study has been useful in the characterization of mobility and physical activity.
- Sedentary behaviors were calculated from the lower back device data. As the position of this device seems to be the cause of some of the complains of the patients, Have the authors considered on using other places for this device? For example, some studies have placed a similar device to the anterior mid-thigh of each patient/subject. This seems to be a comfortable place that allows to obtain data on postures and movements. Are there any reasons why the author preferred the lower back to other positions? According to the authors (page 14) that different monitoring sites provide similar insights into individuals’ mobility, sedentary behavior, physical activity and sleep. So, have they tried other positions for the second device?
We would like to thank you for your questions. Although some participants (from different cohorts including Parkinson’s Disease, Alzheimer’s or Delirium) reported lack of comfort in using the lower back devices, the satisfaction ratio was above 80% among all cohorts in general. However, given that the gait algorithm was developed, validated and reinforced on Lower Back (inverse pendulum), we used this device for more robust outcomes. On the other hand, other algorithms use multi-sensors for calculation of gait parameters which puts more burden on participants (wearing more devices than just one). Besides, other locations (i.e., one device on mid-thigh) cannot precisely calculate gait parameters (using inverted pendulum model) or sedentary behaviours (sitting and lying angles would be similar on thigh). Other discomforts, such as using a band-tape to attach the IMU to thigh would interfere with daily activities such as taking shower and requires more shaving, while attaching the lower back IMU using a double-side tape does not interfere with daily activities.
Regarding the position of the second device, we asked participants to wear it on their wrist for calculation on sleep outcomes, but the GGIR algorithm does not provide gait outcomes like the gait algorithm. By our statement ‘different monitoring sites provide similar insights into individuals’ mobility, sedentary behaviour, physical activity and sleep’ we meant general physical activity patterns and sleep, not the gait macro and micro outcomes. For example, both wrist and lower back devices can provide length of active bouts, but the wrist device cannot provide gait macro and micro outcomes.
- Though the number of participants is low and their ages are limited to the range 47-90, have the authors notice any differences in usability or acceptability depending on the patients’ ages or other conditions? Could the 55% of little to moderate technical issues reported by the patients affect the recorded data? Or the devices’ usability/acceptability? Could these issues be solved or reduced somehow?
Previous studies conducted by corresponding author’s team among 30 people living with Parkinson’s disease portrayed a significant negative correlation between age and the overall scores given to the wearable devices. In this study, we did not find a significant correlation between age and acceptability, which could be due to our low sample size.
Regarding the impacts of technical issue on the recorded data, we would like to declare that it was about participants’ experiences on how to set the device up or re-attach it. It did not impact the recorded data as the device was already setup by the researcher for 7 days with no pause.
- At least in my copy labels in figure 2 are unreadable. The graphs in figure 2 seem to be just low-resolution captures. Please correct. The same for figure 4, labels are unreadable.
Thanks for noticing this. Quality of figures was automatically decreased by word. We attached figures with highest quality to the materials. Their quality will be high in the final publication.
- At least in my copy labels in figure 3b are difficult/uncomfortable to read. Can this be solved?
Quality of figures was automatically decreased by word. We attached figures with highest quality to the materials. Their quality will be high in the final publication.
- If the study downsampling is aimed at analyzing the possibility of extend monitoring of individuals with RA, what are the sizes of the recorded files measuring at 100Hz? And what is the storage capacity of the used devices? Of course, probably no fast movements requiring high recording frequencies would probably expected, but perhaps using high frequencies (100 Hz) could extend the aim of the use of these devices to detection of falls or similar conditions, if they’re interesting.
Thanks for your suggestion. The total size of both 3D acceleration and gyroscope (100Hz) signals for 7 days was around 740MB and the total storage capacity of the IMU was 1GB. The information on the Axivity devices is on their website so we decided not to re-report them in this manuscript.
Since the ratio of freezing of gait and risk of fall was not the main aim of this study, we did not analyse them. We would like to thank you for your suggestion. We will consider it for our future studies.
Reviewer 3 Report
Comments and Suggestions for Authors
The article makes a significant contribution to the applicability of wearable devices for RA patients. Overall, it is well-written and structured. Although the sample size is small, I don't believe this undermines the statistical validity of the PCA analysis.
I have just a few suggestions:
In the abstract, avoid including abbreviations for terms that are not used later in the text.
Please double-check Figure 2.
Avoid using words from the title as keywords.
The discussion section needs to be expanded. There are numerous diagnostic systems where wearable technology has been applied. You could incorporate their findings into your discussion section.
In the limitations section, I suggest adding a few details about the wearability of the device.
While I recommend increasing the sample size, I understand this might be challenging at this stage of the study.
Author Response
Dear respected reviewer,
Hello and thank you for your constructive comments. We tried to amend the manuscript in line with your comments and suggestions. Please find our responses below:
- In the abstract, avoid including abbreviations for terms that are not used later in the text.
Thank you for your comment. It was amended.
- Please double-check Figure 2.
We checked and amended the figure.
- Avoid using words from the title as keywords.
Thank you for your comment. We amended the keywords.
- The discussion section needs to be expanded. There are numerous diagnostic systems where wearable technology has been applied. You could incorporate their findings into your discussion section.
Thank you for your feedback. We have expanded the discussion while maintaining coherence with RA in the overall structure.
- In the limitations section, I suggest adding a few details about the wearability of the device.
Thanks for your suggestion. We added it to the lines 414-416.
- While I recommend increasing the sample size, I understand this might be challenging at this stage of the study.
Thank you for your suggestion. As this was a pilot study funded for a duration of one year, we were unable to recruit additional participants with RA. Consequently, increasing the sample size at this stage is not feasible.
We would also like to emphasize that, despite the small sample size, the study involved 11 participants over 4 sessions, each lasting 7 days (resulting in a total of 44 assessments over 308 days). This level of data collection is consistent with the scope and objectives of a pilot study and supports the acceptability of our outcomes.